# In Vitro and In Vivo Human Body Odor Analysis Method Using GO:PANI/ZNRs/ZIF−8 Adsorbent Followed by GC/MS

**DOI:** 10.3390/molecules27154795

**Published:** 2022-07-27

**Authors:** Sehyun Kim, Sunyoung Bae

**Affiliations:** Department of Chemistry, Seoul Women’s University, 621 Hwarang-ro, Nowon-gu, Seoul 01797, Korea; shgiwork@naver.com

**Keywords:** human body odor, volatile organic compounds, GO:PANI/ZNRs/ZIF−8, response surface methodology, needle-based adsorbent, headspace-in needle microextraction

## Abstract

Among various volatile organic compounds (VOCs) emitted from human skin, *trans*-2-nonenal, benzothiazole, hexyl salicylate, α-hexyl cinnamaldehyde, and isopropyl palmitate are key indicators associated with the degrees of aging. In our study, extraction and determination methods of human body odor are newly developed using headspace-in needle microextraction (HS-INME). The adsorbent was synthesized with graphene oxide:polyaniline/zinc nanorods/zeolitic imidazolate framework-8 (GO:PANI/ZNRs/ZIF−8). Then, a wire coated with the adsorbent was placed into the adsorption kit to be directly exposed to human skin as in vivo sampling and inserted into the needle so that it was able to be desorbed at the GC injector. The adsorption kit was made in-house with a 3D printer. For the in vitro method, the wire coated with the adsorbent was inserted into the needle and exposed to the headspace of the vial. When a cotton T-shirt containing body odor was transferred to a vial, the headspace of the vial was saturated with body odor VOCs. After volatile organic compounds were adsorbed in the dynamic mode, the needle was transferred to the injector for analysis of the volatile organic compounds by gas chromatography/mass spectrometry (GC/MS). The conditions of adsorbent fabrication and extraction for body odor compounds were optimized by response surface methodology (RSM). In conclusion, it was able to synthesize GO:PANI/ZNRs/ZIF−8 at the optimal condition and applicable to both in vivo and in vitro methods for body odor VOCs analysis.

## 1. Introduction

Human body odor is a mixture of compounds generated by skin bacteria resulting in volatile organic compounds (VOCs) such as alcohol, aldehyde, acid, amine, hydrocarbons, and so on [1,2,3,4,5,6]. The primary body odor, which is distinguishable from others, is very unique in individuals that vary with age, gender, ethnic, body parts, and so on [1,5]. It can be important in areas of medical, security, safety, and cosmetic applications. On the basis of the literature, body odor VOCs carry age-related information and can be analyzed by various methods [1,6,7,8,9]. Previous studies have shown that *trans*-2-nonenal, benzothiazole, and isopropyl palmitate are commonly found in the elderly [1,7,8], while hexyl salicylate and α-hexyl cinnamaldehyde occur more frequently in young people [7]. In particular, *trans*-2-nonenal is known to be secreted more as ω7 unsaturated fatty acids are oxidized due to the increase of ω7 unsaturated fatty acids and lipid peroxides as skin aging progresses [1]. In addition, reactive oxygen species (ROS) generated by ultraviolet rays may also react with unsaturated fatty acids to affect the generation of body odor [9].

Organic solvents, gauze, solid phase microextraction (SPME) fiber, and stir bar have been used to extract human body odor [6,10,11,12]. The SPME method has been commonly used to avoid using organic solvents. There are two methods for analyzing body odor with SPME: in vivo and in vitro [13]. In both methods, target compounds are absorbed onto SPME fiber and then thermally desorbed and analyzed by gas chromatography/mass spectrometry (GC/MS). In the in vitro method, a cotton swab or T-shirt that has absorbed body odor is transferred to a SPME vial and extracted with SPME fiber. The in vivo method is a method to directly extract body odor by installing the SPME device on the skin [13]. However, it is quite inconvenient because of a glass funnel that fasten the SPME device on the skin. To address these limitations, a new method of body odor analysis has been developed by in-needle microextraction (INME) using graphene oxide:polyaniline/zinc nanorods/zeolitic imidazolate framework-8 (GO:PANI/ZNRs/ZIF−8) adsorbent. The in-needle microextraction (INME) technique has been used for extraction of volatile organic compounds in various matrix due to easy fabrication and cost effectiveness [14,15,16,17,18,19,20]. The adsorbent has been synthesized with various polarity based on polyaniline modified by additives such as multiwall carbon nanotube and ionic liquid [14,19]. The adsorbent containing analytes was desorbed by gas chromatograph/mass spectrometer (GC/MS) for qualitative and quantitative analyses.

INME has an adsorbent inside the needle and is used for solid dynamic microextraction. INME uses the needle coated with the adsorbent inside or the wire coated with adsorbent being inserted inside of the needle for extraction and analysis without using solvents followed by chromatographic analysis [14,15,16,17,18,19,20]. Especially, dynamic headspace-INME (HS-INME) has been applied to the sample by exposing the INME needle with a pump to the headspace of the vial saturated with VOCs. Polymeric adsorbents have been synthesized by sol–gel reaction [15,17,19] and/or electrochemical deposition methods [14,16,18]. It has demonstrated that INME could be customized due to easy fabrication of adsorbent and economically feasible to extract target compounds without using solvents.

The objective of this study is to fabricate GO:PANI:ZNRs:ZIF−8 adsorbent and utilize this for direct extraction of the body odor from human skin in vivo and for indirect extraction from the media adsorbed with body odor in vitro, followed by GC/MS. The mixing ratio of ZNRs and ZIF−8 of the adsorbent was varied to optimize the adsorption by modifying polarity [21]. Porous GO:PANI can be easily coated onto stainless-steel wire from the electrochemical method, while ZNRs exhibit good affinity with polar compounds and ZIF−8 with polar compounds [21,22]. Characterization of the adsorbent was measured by Fourier-transform infrared spectroscopy (FTIR), Thermogravimetric Analyzer (TGA), X-ray diffractometer (XRD), Brunauer–Emmett–Teller (BET) surface area method, and field emission-scanning electron microscopy (FE-SEM). Fabrication and extraction conditions were optimized by the Box-Behnken design (BBD) combined with response surface methodology (RSM), and the developed method was validated including limit of detection (LOD), limit of quantification (LOQ), recovery, and reproducibility.

## 2. Materials and Methods

### 2.1. Chemicals and Instrumentation

Five target compounds (Appendix A), *trans*-2-nonenal (97%), benzothiazole (96%), hexyl salicylate (99%), α-hexyl cinnamaldehyde (85%), and isopropyl palmitate (90%), were obtained from Sigma-Aldrich (St. Louis, MO, USA). Stock solution of all target compounds was prepared using methanol (HPLC grade, Samchun Pure Chemical Co., Pyeongtaek, Korea) at a concentration of 1000 mg L^−1^. Stainless-steel wire (0.22 mm O.D., 09 one Science, Goyang, Korea) was used as an adsorbent wire and a working electrode. A Hamilton 91022 needle (metal hub, 22 gauge, 718 μm O.D., 413 μm I.D., bevel tip, 51 mm length) and 1 mL Hamilton 1001 TLL (gas tight syringe, luer lock gas tight syringe barrel, Reno, NV, USA) with plunger (polytetrafluoroethylene, Reno, NV, USA) were used for INME adsorption and GC/MS thermal desorption. Chemicals used in fabrication of adsorbent was shown in the Appendix A.

Agilent 6970 gas chromatograph (GC) with a 5973 mass selective detector (MS) was used to analyze target compounds. HP-5MS (30 m × 0.25 mm × 0.25 μm, (5%-phenyl)-methylpolysiloxane) was used as the column and helium with the flow rate of 1 mL min^−1^ as the carrier gas. The injector temperature was 240 °C using split mode (30:1), and the oven program was 60 °C (4 min) to 230 °C (10 min) at 6 °C min^−1^. The range of 50~350 *m*/*z* was scanned in full scan mode.

Adsorbent fixing part and cover of the adsorption kit were 3D printed on a Cubicon Single Plus (3DP-310F) with polylactic acid (PLA)-plus filament.

### 2.2. Fabrication of Wire Coated with Adsorbent

The adsorbent was prepared in 3 steps: (1) fabrication of GO:PANI layer, (2) fabrication of GO:PANI/ZNRs layer, and (3) fabrication of GO:PANI/ZNRs/ZIF−8 layer, as illustrated in Figure 1.

GO:PANI layer was fabricated on stainless-steel wire using cyclic voltammetry in which 0.6–1.0 V (sweeping rate of 50 mV s^−1^ for 30 cycles) was applied to an electrolyte containing GO and aniline in a H_2_SO_4_ (0.5 mol L^−1^) solvent. Three-electrode system consisting of stainless-steel wire (working electrode), Pt wire (counter electrode), and Ag/AgCl electrode (reference electrode) was used. After electrochemical deposition, the coating layer was rinsed several times using distilled water to remove unreacted chemicals. Finally, the stainless-steel wire coated with porous GO:PANI was dried at 80 °C for 0.5 h.

In the GO:PANI/ZNR fabrication step, the GO:PANI coated stainless-steel wire was dipping into an aqueous mixture of zinc nitrate hexahydrate (0.5 mol L^−1^) and hexamethylenetetramine (0.5 mol L^−1^) for 30 s and dried in 180 °C oven for 2 min. This step was repeated five times. In this process, ZnO seeds formed by OH^−^ and Zn^2+^ were deposited on the porous GO:PANI surface. Then, the stainless-steel wire was dipped into the aqueous mixture of zinc nitrate hexahydrate (0.05 mol L^−1^) and hexamethylenetetramine (0.05 mol L^−1^) and placed in 95 °C oven for 2 h to hydrothermally grow ZnO nanorods [22].

Finally, the stainless-steel wire coated with GO:PANI/ZNRs was dipped into a 2-MI solution (DMF:H_2_O = 3:1 (*v*/*v*)) and placed in 90 °C oven for 8 h [22]. In this procedure, ZIF−8 was formed by coordination bonding between 2-MI and Zn^2+^ on ZNRs surface. Prior to the extraction process, the adsorbent coated on stainless-steel wire was placed in a 220 °C oven for 1 h to remove impurities from the coated wires. Then, it was kept in a desiccator at room temperature until further use.

### 2.3. Characterization of GO:PANI/ZNRs/ZIF−8 Adsorbent

The functional groups of the adsorbent generated at each step to confirm the synthesis were measured by the Fourier-transform infrared (FT-IR) spectroscopy (Perkin Elmer Spectrometer 100, Waltham, MA, USA). The crystallinity of synthesized graphene oxide and coating layer of each synthesis step were confirmed using X-ray diffractometer (XRD, Bruker DE/D8 Advance, Bruker, Karlsruhe, Germany). The thermal stability of GO:PANI, GO:PANI/ZNRs, and GO:PANI/ZNRs/ZIF−8 were evaluated by thermogravimetric analyzer (TGA, TG209 F1 Libra, Netzsch, Selb, Germany).

The specific surface area, pore size distribution, and N_2_ adsorption-desorption isotherm of GO:PANI/ZNRs and GO:PANI/ZNRs/ZIF−8 were observed by the Brunauer–Emmet–Teller (BET, 3flex, Micromeritics, Norcross, GA, USA). The morphology of GO:PANI, GO:PANI/ZNRs, and GO:PANI/ZNRs/ZIF−8 was characterized using field emission-scanning electron microscope (FE-SEM, JSM-6700F, JEOL Ltd., Tokyo, Japan).

### 2.4. INME Process

As shown in Figure 2, extraction of the target compounds was performed by the INME method. In the preparation of the INME adsorption device, a GO:PANI/ZNRs/ZIF−8-coated stainless-steel wire was inserted vertically into a Hamilton needle and then connected to a gas tight syringe. After spiking the target compounds into the SPME vial, it was sealed with a mini-nut cap. Then, saturation was performed at 40 °C for 15 min. Next, dynamic adsorption was carried out by penetrating the needle to the headspace of the SPME vial. A homemade reciprocating pump was used at 6 cycles/min to suck the analytes saturated in the headspace inside the needle during this process. Upon completion of adsorption, the analytes were thermally desorbed from the adsorbent by placing it in a GC/MS injector and analyzed by GC/MS.

### 2.5. Adsorption Kit Fabrication

The body odor adsorption kit was designed to adsorb VOCs directly from the skin (Figure 3). It consists of adsorbent, adsorbent fixing part, cover, and medical tape. The adsorbent was coated on the middle section of 5.0 cm stainless-steel wire to be placed in the fixing part. An adsorbent fixing part and its cover were manufactured by a 3D printer with PLA plus filament. The adsorbent kit was made as a cylinder with a hollow in the middle and a closed top, which could collect VOCs from the skin and protect the adsorbent from external odors and impacts. In addition, the adsorbent was about 10 mm away from the skin, which could avoid contamination from dead skin cells and sweat. The adsorbent fixing part has a hole in the center of the top to insert the adsorbent. PDMS was applied on the bottom of the adsorbent fixing part to reduce skin irritation due to direct contact on skin. The adsorbent kit was assembled to conduct the experiments; insert the adsorbent into the fixing part, close the cover, put it on the skin, and attach the medical tape to fix it to the skin. After the body odor was collected for 2 h, the wire coated with adsorbent was removed from the kit, inserted into a Hamilton needle, connected to a gas tight syringe, thermally desorbed at GC/MS inlet, and the body odor components were analyzed in GC/MS system. For the feasibility test of the adsorption kit, the target compounds were spiked to the artificial silica skin, and the adsorption kit was assembled to place it and adsorbed for 2 h at 36 °C.

### 2.6. Optimization Using Response Surface Methodology (RSM)

In the fabrication of adsorbent, the concentrations of GO, aniline, and 2-MI that may affect adsorption amount of the target compounds were optimized using RSM. 3-factor 3-level Box BBD was used with Minitab 19 (Minitab Inc., State College, PA, USA). A total of 17 experiments were conducted randomly, as shown in Appendix A. The levels and coded numbers of each factor are as follows: amount of GO (X_1_) was 5 mg L^−1^ (−1), 10 mg L^−1^ (0), and 15 mg L^−1^ (1); amount of aniline (X_2_) was 9.31 g L^−1^ (−1), 14.0 g L^−1^ (0), and 18.6 g L^−1^ (1); and amount of 2-MI (X_3_) was 10.26 g L^−1^ (−1), 102.6 g L^−1^ (0), and 195.0 g L^−1^ (1).

In addition, the 3-factor 3-level BBD was used to optimize the INME analysis condition. As shown in Appendix A, total of 17 experiments were conducted randomly. Extraction temperature (X_1_), adsorption time (X_2_), and desorption time (X_3_) were selected as factors. The levels and coded numbers for each factor were as follows: extraction temperature (X_1_) was 30 °C (−1), 40 °C (0), and 50 °C (1); the adsorption time (X_2_) was 30 min (−1), 45 min (0), and 60 min (1); and the desorption time (X_3_) was 1 min (−1), 3 min (0), and 5 min (1).

The data obtained from the experiments for fabrication condition and the INME analysis conditions were analyzed by the response surface regression procedure using second-order polynomial model (Equation (1)).
(1)$$Y=\alpha_{0}+\alpha_{1}X_{1}+\alpha_{2}X_{2}+\alpha_{3}X_{3}+\alpha_{11}X_{1}^{2}+\alpha_{22}X_{2}^{2}+\alpha_{33}X_{3}^{2}+\alpha_{12}X_{1}X_{2}+\alpha_{13}X_{1}X_{3}+\alpha_{23}X_{2}X_{3}$$where Y is the predicted response (amount of adsorption); $\alpha_{0}$ is the intercept, $\alpha_{1}$, $\alpha_{2}$, and $\alpha_{3}$ are linear coefficients; $\alpha_{11}$, $\alpha_{22}$, and $\alpha_{33}$ are quadratic coefficients; and $\alpha_{12}$, $\alpha_{13}$, and $\alpha_{23}$, are the interactive coefficients.

## 3. Results and Discussion

### 3.1. Optimization of Human Body Odor Adsorbent Fabrication Conditions Using RSM

Reactants such as GO (X_1_), aniline (X_2_), and 2-MI (X_3_) added were optimized through RSM. It determines the optimized value of each variable and analyzes the interactions between the variables. Optimized values for each variable were obtained using the regression equation calculated using Minitab (Table 1). A three-dimensional response surface curve is a graphical representation of the results from the regression equation. In each graph, after fixing one of the three independent variables to level 0, the effect of the remaining two independent variables on peak area obtained from the total ion chromatogram (TIC) was investigated (Appendix A) [23]. Confirming the relationship between GO and aniline, the peak area value rapidly increased as aniline increased, while the peak area value increased gently as the amount of GO increased. In the relationship between GO and 2-MI, as the amount of 2-MI increased, the peak area increased rapidly except for *trans*-2-nonenal, and the amount of GO showed a gradual increase. In the relationship between aniline and 2-MI, the peak area increased rapidly as 2-MI increased, except for *trans*-2-nonenal, and the amount of aniline increased gradually. Consequently, the order affecting the peak areas is 2-MI, aniline, and GO. The optimal BBD results for the reactants are determined as 15 mg L^−1^ of GO, 18.3 g L^−1^ of aniline, and 151.6 g L^−1^ of 2-MI. The error of the modeling equation obtained through the BBD method was about 10%. The peak area value for each target compound can be predicted by substituting the concentration of each reactant.

### 3.2. Optimization of INME Extraction Conditions Using RSM

HS-INME-GO:PANI/ZNRs/ZIF−8 extraction conditions, including extraction temperature (X_1_), adsorption time (X_2_), and desorption time (X_3_), were optimized using the RSM method. Appendix A is a response surface graph showing the relationship between the variables and the peak area values. In the relationship of extraction temperature and adsorption time, it was observed that the extraction temperature had a more significant effect on extraction for *trans*-2-nonenal and benzothiazole, while the adsorption time had a greater effect on the remaining compounds. Confirming the relationship of extraction temperature and desorption time, the extraction temperature had a greater effect than the desorption time for all target compounds. The relationship between adsorption time and desorption time was found that desorption time had a greater effect on extraction. The optimal extraction conditions based on the RSM results are as follows. The optimal condition was 40 °C for extraction temperature, 60 min for adsorption time, and 4.3 min for desorption time, respectively. The modeling equation showed an error about 10%. and the value of the peak area of each target compound could be reasonably predicted from the modeling equation (Table 2).

### 3.3. Characterization of the Adsorbent

The products generated at each synthesis process prepared under optimal conditions were characterized through FT-IR, TGA, XRD, BET, and FE-SEM. FT-IR spectra (Figure 4A) show that GO was well-synthesized from graphite by verifying the O-H band at 3389 cm^−1^, the C=O band at 1721 cm^−1^, the C=C band in the phenol ring at 1613 cm^−1^, the C-OH band at 1227 cm^−1^, and C-O band at 1048 cm^−1^ in the IR spectrum of GO [24]. Through the following peaks in the IR spectrum of GO:PANI, it was confirmed that the GO plate was covered by PANI nanofibers to form GO:PANI identified by peaks at 1561 cm^−1^ from C=C in quinone, 1479 cm^−1^ from benzene ring, 1302 cm^−1^ from C-N in the second amine ring, 1242 cm^−1^ from C-N vibrational band in the protonic acid, 1126 cm^−1^ from C=N vibrational band, and 805 cm^−1^ from the aromatic C-H band [24]. In the IR spectrum of GO:PANI/ZNRs, peaks of amine groups and benzene ring shifted to higher wavenumbers than GO:PANI from the previous synthesis step were observed at 1585 cm^−1^, 1498 cm^−1^, 1297 cm^−1^, 1143 cm^−1^, and 826 cm^−1^. This indicates that the amine and imine nitrogen atoms interact with Zn^2+^ via either protonation or complexation [25]. In addition, the O-Zn-O stretching vibration peak was observed at 432 cm^−1^ [26]. Amine groups and a benzene ring were observed at 1596 cm^−1^, 1500 cm^−1^, 1308 cm^−1^, 1148 cm^−1^, and 827 cm^−1^ in the GO:PANI/ZNRs/ZIF−8 IR spectrum. Additionally, a peak at 748 cm^−1^ from out of plane bending of the 2-MI ring and a 420 cm^−1^ peak from Zn-N stretching confirmed that the nitrogen atom was connected to 2-MI linker to form ZIF−8 [27].

From the XRD analysis, it was confirmed that ZNRs and ZIF−8 were well-formed (Figure 4B). In the GO:PANI/ZNRs, peaks at 2θ = 31.68° (100), 34.35° (002), 36.16° (101), 47.43° (012), and 56.49° (110), which are identical to the peaks from the ZnO standard, were observed. Both ZnO and ZIF−8 peaks in GO:PANI/ZNRs/ZIF−8 were confirmed to compare with peaks from the ZnO and ZIF−8 standards. Peaks observed at 2θ = 7.33° (011), 10.37° (002), 12.71° (112), 14.70° (022), 16.43° (013), 18.52° (222), 22.14° (114), 24.48° (233), and 26.61° (134) were identical to the peaks of the ZIF−8 standard, while peaks at 2θ = 31.71° (100), 34.40° (002), 36.20° (101), 47.48° (012), and 56.53° (110) were observed from the ZnO standard. It was concluded that the adsorbent consists of GO:PANI/ZNRs/ZIF−8 was well-synthesized and expected to adsorb compounds having various polarities due to various functional groups.

Thermal stability of the adsorbent synthesized at each step was measured by TGA (Appendix A). Significant weight loss was not observed and confirmed thermal stability up to 273 °C. Mass change at 81 °C is regarded as the dehydration process. The mass change of GO:PANI/ZNRs was not observed until 262 °C. The effect of GO addition on thermal stability was shown clearly to compare the decomposition temperature between PANI/ZNRs/ZIF−8 and GO:PANI/ZNRs/ZIF−8, which was 195 °C and 291 °C, relatively.

The effect of ZIF−8 addition on pore size distribution is shown in Figure 5 as the pore size distribution and N_2_ adsorption and desorption isotherm. The BET specific surface area of GO:PANI/ZNRs/ZIF−8 was 4640 m^2^ g^−1^, which was much higher than that of GO:PANI/ZNRs with a specific surface area of 5.35 m^2^ g^−1^. According to the pore size distribution curve, GO:PANI/ZNRs has mesopore (2~50 nm), whereas GO:PANI/ZNRs/ZIF−8 has micropore (<2 nm) and mesopore (2~50 nm) [28]. In addition, N_2_ adsorption and desorption isotherm of GO:PANI/ZNRs/ZIF−8 showed type I isotherm indicating that micropore and mesopore coexist [29].

The morphology of the adsorbent coated on stainless-steel wire was observed using FE-SEM in Figure 6. The image of GO:PANI showed a rough and porous surface due to the deposition of PANI nanofibers on the GO sheets [24]. In GO:PANI/ZNRs image, flower-shaped ZnO nanostructures were constructed on the porous GO:PANI. Rhombic dodecahedron ZIF−8 was observed in GO:PANI/ZNRs/ZIF−8 image, which indicates that ZIF−8 was well-dispersed on the ZNRs surface of GO:PANI/ZNRs [30].

Comparing the adsorption amount by each synthesized step and PANI/ZNRs/ZIF−8, GO:PANI/ZNRs/ZIF−8 shows the highest adsorption for all target compounds (Appendix A). The final adsorbent has moieties with different polarities from ZNRs and ZIF−8 to enhance the adsorption potentials, and its large specific surface area and micropores might play an important role in increasing the adsorption amounts. Strong π–π stacking interaction and hydrogen bonding between target compounds and adsorbent contribute to the improved extraction of target compounds [22,23]. In comparison between GO:PANI/ZNRs/ZIF−8 and PANI/ZNRs/ZIF−8, GO addition tends to increase the adsorption amount of *trans*-2-nonenal that appears to interact between GO and C=O of *trans*-2-nonenal.

Based on the literature reviews, there are few studies on GO:PANI/ZNRs/ZIF−8 adsorbent for the human body odor compounds. On the other hand, the adsorbents composed of GO, PANI, ZNRs, and ZIF−8 as SPME fibers coupled with chromatography have been studied in water, soil, air, and food samples [21,22,24,27,28]. The GO:PANI was used as an effective adsorbent for Zn ion removal from the water sample [24]. The adsorbent of zinc oxide nanorods directly coated on stainless-steel wire was applied as SPME fiber to extract aldehydes for instant noodle samples [21]. Hexanal, heptanal, octanal, nonanal, and decanal were analyzed with a linear range between 0.08 and 5.0 μg g^−1^ [21]. PANI/ZNRs/ZIF−8 was synthesized to extract 2-methylnaphthalene, 1-methylnaphthalene, pyrene, phenanthrene, cyrysene, and benzo[a]pyrene of polycyclic aromatic hydrocarbons (PAHs) in a sewage water sample [22]. The synthesized ZIFs-ZnO composite as a sorbent for SPME-HPLC-UV was used to remove Sudan dyes in the water sample showing 0.002 ng mL^−1^ LOD and wide linear ranges 0.02–20 ng mL^−1^ [27].

### 3.4. Comparison of Extraction Efficiency

The Dynamic INME and static INME method were compared by enrichment factor (EF). EF was calculated as follows:EF = A_1_/A_0_(2)

A_1_ is the peak area from dynamic INME method that sucks the target compounds inside the needle using a reciprocating pump, and A_0_ is the peak are from the static INME method that was performed without using a pump. The EF value of *trans*-2-nonenal was 1062.29 ± 33.72, benzothiazole was 390.12 ± 73.14, hexyl salicylate was 29.71 ± 5.41, α-hexyl cinnamaldehyde was 36.79 ± 1.30, and isopropyl palmitate was 1.54 ± 0.59, respectively. It was concluded that all target compounds were more efficiently extracted than the static INME- method.

### 3.5. Feasibility Test Using an Adsorption Kit

The body odor adsorption kit developed in this study was applied to in vivo sampling. A feasibility test was performed on petri dish spiking 30 μg target compounds to place an adsorption kit fixed with the medical tape.

After 2 h exposure to compounds, the wire coated with GO:PANI/ZNRs/ZIF−8 adsorbent inserted inside the adsorption kit was removed, put into the Hamilton needle, and connected to the plunger to desorb the compounds at GC/MS injector. The TIC obtained from this process is shown in Figure 7. From TIC, all five target compounds were well-separated and measured. *trans*-2-Nonenal, benzothiazole, hexyl salicylate, α-hexyl cinnamaldehyde, and isopropyl palmitate were successfully separated at the retention time of 12.45 min, 14.42 min, 24.15 min, 25.40 min, and 29.92 min, respectively. In conclusion, the body odor adsorption kit developed in this study could be used as an in vivo sampling method for human body odor analysis.

### 3.6. Method Validation

The INME method for in vitro body odor extraction was validated by analyzing five target compounds repeatedly. For all target compounds, the linearity of the seven-points calibration curves had r^2^ values greater than 0.99 in the dynamic range. Limit of detection (LOD) and limit of quantitation (LOQ) were calculated according to ISO definition [31]. The calculated LOD and LOQ were 4.89 ng~128 ng and 14.8 ng~38.8 ng, respectively. The dynamic range was 4.89 ng~15,000 ng.
Recovery = (A_C_ + A_I_) − A_0_/A_S_ × 100(3)

In addition, a recovery test was performed by analyzing a 100% cotton T-shirt with an analyte spiked by the proposed method (Figure 8). The recovery rate was calculated using Equation (3), that A_C_ is the amount of target compounds adsorbed on the 100% cotton T-shirt, A_I_ is the amount of target compounds adsorbed on GO:PANI/ZNRs/ZIF−8 adsorbent using HS-INME method, A_0_ is the amount of target compounds in the unspiked sample, and A_S_ is the amount of target compounds spiked on the 100% cotton T-shirt [19]. The recovery rate shows acceptable values, ranging from 91.27% to 103.47%.

The reproducibility of the proposed method was confirmed by obtaining the relative standard deviation of intra assay (run to run, *n* = 5) and inter assay (needle to needle, *n* = 5). As a result, in the intra and inter assays, all target compounds showed about 10% and 15% precision, respectively.

## 4. Conclusions

In this study, in vitro and in vivo methods using GO:PANI/ZNRs/ZIF−8 adsorbent that were suitable for body odor adsorption was developed. The amounts of GO, aniline, and 2-MI added during the adsorbent fabrication process were simply and accurately optimized through the RSM method determined as 15 mg L^−1^ of GO, 18.3 g L^−1^ of aniline, and 151.6 g L^−1^ of 2-MI, respectively. The physicochemical characterization of the synthesized adsorbent was confirmed by FT-IR, XRD, TGA, BET, and SEM. The large surface area and micropore of the adsorbent increased the adsorption amount of the target compounds, and the thermal stability was increased due to the addition of GO. In in vivo sampling, a wire coated with adsorbent could be placed directly to the human skin using an adsorbent kit developed in this study. In in vitro method, a wire coated with GO:PANI/ZNRs/ZIF−8 to be used as dynamic HS-INME method to adsorb human body odor VOCs saturated in the headspace of the vial. As a result of the validation of the in vitro method, analysis of aging-related human body odor compounds was successfully performed with good precision and recovery. GO:PANI/ZNRs/ZIF−8 adsorbent could be used as INME method that could adsorb five aging-related human body odor compounds successfully and demonstrate over 100 applications after the conditioning process.

## Figures and Tables

**Figure 1 molecules-27-04795-f001:**
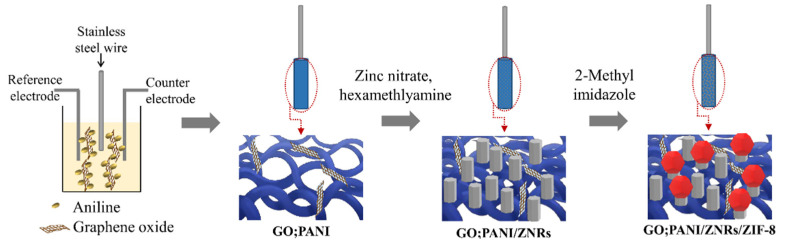
Fabrication process of the adsorbent.

**Figure 2 molecules-27-04795-f002:**
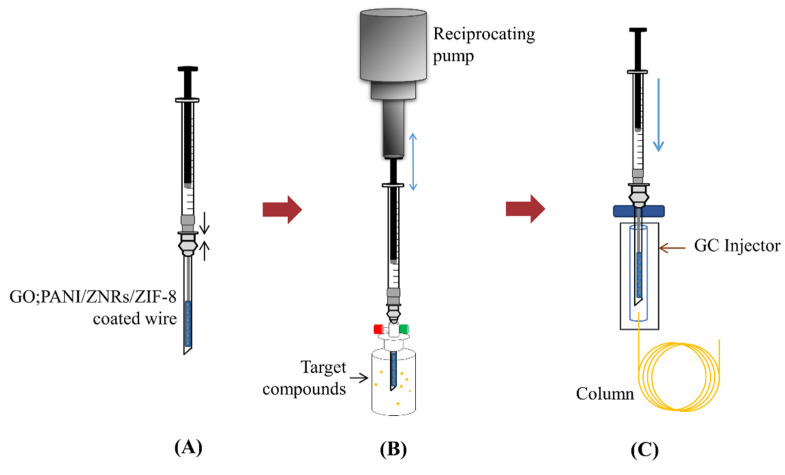
A scheme of INME process. (**A**) A needle inserted with a coated wire is connected to a gas tight syringe. (**B**) The needle was plugged into the mini-nut cap to adsorb target compounds in the headspace using a reciprocating pump. (**C**) The target compounds adsorbed on the adsorbent was thermally desorbed in GC injector and analyze using GC/MS.

**Figure 3 molecules-27-04795-f003:**
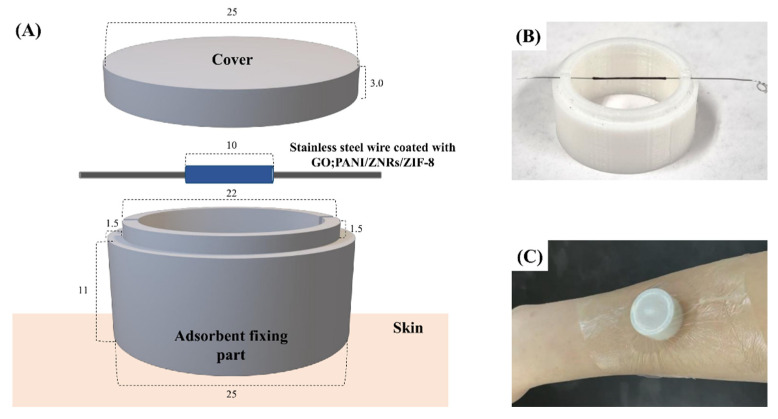
(**A**) A schematic view of adsorption kit for the collection of body odor from human skin (the unit of length: mm). (**B**) Actual image of adsorbent placed on the fixing part made by 3D printer. (**C**) Sampling of body odor at the arm using adsorbent kit covered by a medical tape.

**Figure 4 molecules-27-04795-f004:**
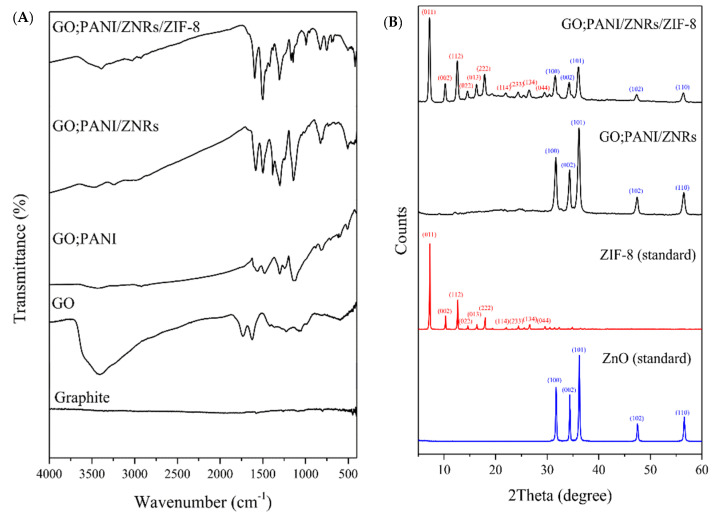
(**A**) FT-IR spectrums of graphite, GO, GO:PANI, GO:PANI/ZNRs, and GO:PANI/ZNRs/ZIF−8 and (**B**) XRD spectrums of the ZnO (standard), ZIF−8 (standard), GO:PANI/ZNRs/ZIF−8, and GO:PANI/ZNRs/ZIF−8.

**Figure 5 molecules-27-04795-f005:**
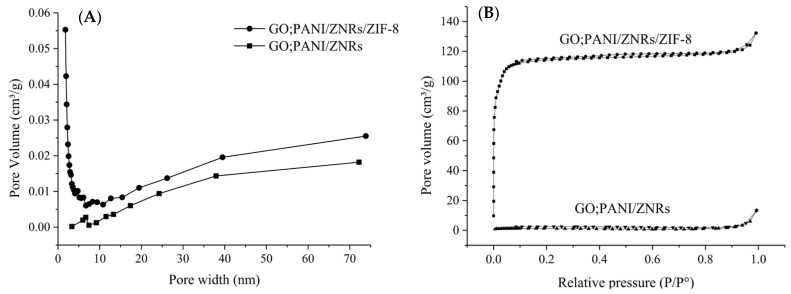
(**A**) Pore width distribution of GO:PANI/ZNRs and GO:PANI/ZNRs/ZIF−8 and (**B**) N_2_ adsorption-desorption isotherms of GO:PANI/ZNRs and GO:PANI/ ZNRs/ZIF−8.

**Figure 6 molecules-27-04795-f006:**
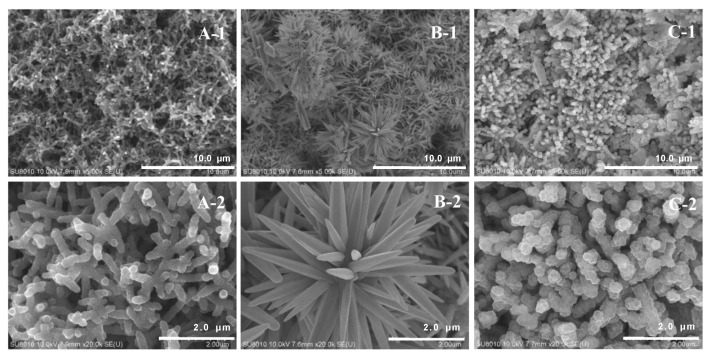
SEM images of stainless-steel wire coated with (**A**) GO:PANI, (**B**) GO:PANI/ZNRs, and (**C**) GO:PANI/ZNRs/ZIF−8 (1; ×5000, 2; ×20,000).

**Figure 7 molecules-27-04795-f007:**
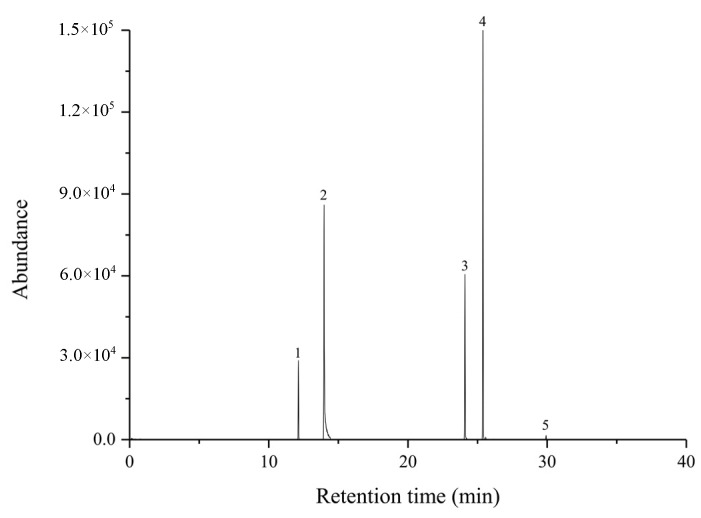
Total ion chromatogram of aging-related body odor using adsorption kit ((1) *trans*-2-noenal, (2) benzothiazole, (3) hexyl salicylate, (4) a-hexyl cinnamaldehyde, and (5) isopropyl palmitate).

**Figure 8 molecules-27-04795-f008:**
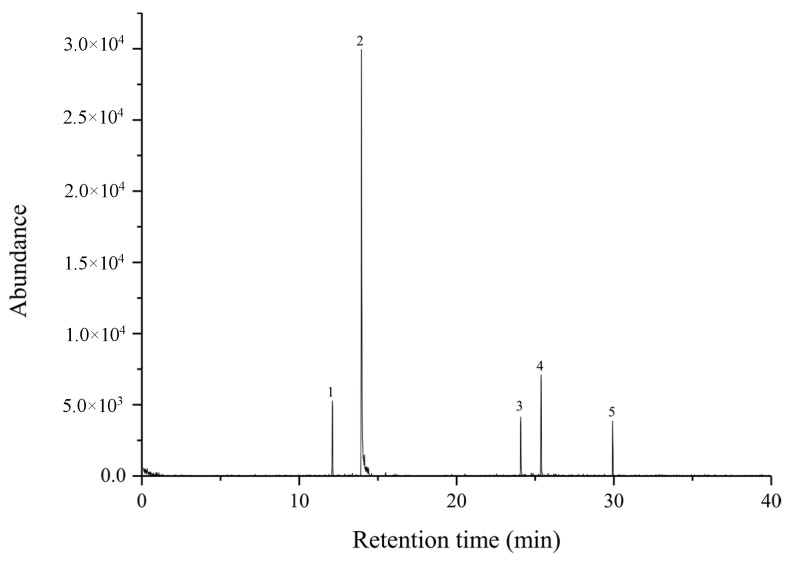
Total ion chromatogram of aging-related body odor spiked on 100% cotton T-shirt using INME method ((1) *trans*-2-nonenal, (2) benzothiazole, (3) hexyl salicylate, (4) a-hexyl cinnamaldehyde, and (5) isopropyl palmitate).

**Table 1 molecules-27-04795-t001:** Peak area prediction formula of each material obtained through BBD in optimization of the reactant concentration.

Compound	Model Formula
*trans*-2-Nonenal	1,373,897 + 65,796 X_1_ + 71,554 X_2_ − 165,521 X_3_ + 737,146 X_1_^2^ − 307,202 X_2_^2^ + 2692 X_3_^2^ − 9779 X_1_X_2_ + 70,330 X_1_X_3_ − 3443 X_2_X_3_
Benzothiazole	2,738,952 + 459,936 X_1_ + 881,137 X_2_ + 255,043 X_3_ + 1,078,534 X_1_^2^ + 520,184 X_2_^2^ + 591,432 X_3_^2^ − 221,829 X_1_X_2_ − 241,702 X_1_X_3_ + 320,824 X_2_X_3_
Isopropyl palmitate	55,849 + 10,191 X_1_ + 10,199 X_2_ + 7497 X_3_ − 11,547 X_1_^2^ − 4503 X_2_^2^ − 15,959 X_3_^2^ + 3465 X_1_X_2_ + 1502 X_1_X_3_ + 3925 X_2_X_3_
Hexyl salicylate	667,682 + 81,671 X_1_ + 81,140 X_2_ + 93,743 X_3_ + 47,534 X_1_^2^ − 18,836 X_2_^2^ − 27,142 X_3_^2^ − 888 X_1_X_2_ + 24,879 X_1_X_3_ + 34,062 X_2_X_3_
α-Hexyl cinnamaldehyde	1,127,759 + 173,707 X_1_ + 296,075 X_2_ + 229,720 X_3_ + 53,389 X_1_^2^ + 84,526 X_2_^2^ + 26,918 X_3_^2^ + 150,333 X_1_X_2_ − 60,256 X_1_X_3_ + 112,567 X_2_X_3_

X_1_: amount of graphene oxide (mg L^−1^), X_2_: amount of aniline (g L^−1^), and amount of X_3_: 2-methyl imidazole (g L^−1^).

**Table 2 molecules-27-04795-t002:** Peak area prediction formula of each compound obtained through BBD in optimization of INME-GO:PANI/ZNRs/ZIF−8 analysis conditions.

Compound	Model Formula
*trans*-2-Nonenal	6,715,422 + 1,488,188 X_1_ + 990,247 X_2_ − 290,098 X_3_ − 154,172 X_1_^2^ − 416,644 X_2_^2^ + 858,182 X_3_^2^ − 1,371,651 X_1_X_2_ + 379,903 X_1_X_3_ − 108,883 X_2_X_3_
Benzothiazole	3,475,053 − 572,871 X_1_ + 578,351 X_2_ + 260,594 X_3_ + 80,741 X_1_^2^ + 19,364X_2_^2^ + 507,335 X_3_^2^ − 1,048,215 X_1_X_2_ − 390,972 X_1_X_3_ + 686,615 X_2_X_3_
Isopropyl palmitate	70,372 + 50,455 X_1_ + 25,337 X_2_ − 1915 X_3_ + 15,132 X_1_^2^ − 7361 X_2_^2^ + 20,237 X_3_^2^ + 9741 X_1_X_2_ − 6146 X_1_X_3_ − 18,574 X_2_X_3_
Hexyl salicylate	1,477,657 + 311,065 X_1_ + 284,017 X_2_ + 104,003 X_3_ − 37,787 X_1_^2^ − 246,864 X_2_^2^ + 85,773 X_3_^2^ + 5355 X_1_X_2_ + 18,324 X_1_X_3_ + 76,669 X_2_X_3_
α-Hexyl cinnamaldehyde	2,459,037 + 6,365,250 X_1_ + 452,391 X_2_ − 67,492 X_3_ − 291,829 X_1_^2^ − 446,990 X_2_^2^ + 118,680 X_3_^2^ + 20,695 X_1_X_2_ − 27,709 X_1_X_3_ + 195,394 X_2_X_3_

X_1_: extraction temperature (°C), X_2_: adsorption time (min), and X_3_: desorption time (min).

## Data Availability

Not applicable.

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
