# Peer review of "In Vitro and In Vivo Human Body Odor Analysis Method Using GO:PANI/ZNRs/ZIF−8 Adsorbent Followed by GC/MS"

_molecules, 2022, doi:10.3390/molecules27154795_

Round 1

Reviewer 1 Report

Authors developed an in-vitro and in-vivo body odor analysis methods for body odors of trans-2-non-enal, benzothiazole, hexyl salicylate, α-hexyl cinnamaldehyde, and isopropyl palmitate. The sample gas material was analyzed by gas chromatography/mass spectrometry. Authors developed adsorption band for adsorbing body odor from human skin, and to and analyze body odor using the developed adsorbent, and applicable to both in-vitro and in-vivo methods. Evaluation of odor analysis sytem, experiments and results are well-written. It is interesting to compare and discuss. Revisions and comments on some of the papers are shown below.

1. How many times can an adsorption device be used for human gas analysis?
2. In Figure 6, author should add a scale bar in SEM images.
3. Authors said “Adsorption band can adsorb five aging related body odor compounds successfully” in conclusion. I think it would be good to present the data of the five comparisons in a table.

Author Response

We greatly appreciate the reviewers for their thoughtful review comments that helped us improve the quality of our manuscript. We have tried our best to thoroughly revise our manuscript by addressing the questions raised by the reviewers. Our detailed responses to the reviewers’ comments were prepared as a separate file.

Reviewer 2 Report

- Title should say what body organism? human or animal etc.
-L8-10 - Rewrite, it is not clear.
- Abstract lack of hypothesis, what is the intention to carry this study, should include.
- L21 should be placed where the experimental design includes, not at the bottom.
- Is it necessary to use VOC as an indicator of aging? as aged people could easily be seen by their appearance. L37-39, revise it. Add another strong reason for this study, instead of the aging factor as the main.
- Introduction is lack reasoning on using GO;PANI/ZNRs/ZIF-8 absorbent, please add more details to support.
- Hypothesis is missing in the introduction part, it looks like the introduction has no end, please modify it.
- is GC condition and Absorbent fabrication author own, or followed any references, if so please specify.
- What is the purpose of RSM in this study?
- L172 - is 2 h sufficient for absorption, is there any temperature maintained? if so please include it.
- Is the material and method section complete? I have not found any characterization of absorbent determination in this section. !!!
- Include statistical procedures.
- Results and discussion has limited comparative details, the author should include details from literature to exhibit a connection or explain more about how this absorbent is efficient, and so on. 
- As the authors used human samples, are there any ethical problems? please include if the author gets permission to work on human samples. 

Author Response

(The authors gave the same response as above.)

Reviewer 3 Report

In this paper, the GO;PANI/ZNRs/ZIF-8 absorbent were developed and applied for the analysis of in-vitro and in-vivo body odor indicators (trans-2-nonenal, benzothiazole, hexyl salicylate, α-hexyl cinnamaldehyde, and isopropyl palmitate) combined with GC/MS. The amounts of GO, aniline, and 2-MI added in the fabrication of adsorbent were optimized by the RSM method. The synthesized adsorbent was confirmed via FT-IR, 406 XRD, TGA, BET, and SEM. I suggest publishing this manuscript after a minor reversion.

1. The purity of some purchased target compounds [(α-hexyl cinnamaldehyde (85%) and isopropyl palmitate (90%)] is low, which may affect the analysis of these compounds.

2. The oven procedure is 60-230 ℃ in GC-MS detection, but the boiling point of Hexyl salicylate is 290 ℃, how to determine this compound?

3. It is recommended to reprocess the FT-IR spectrums and flatten the baseline.

Author Response

(The authors gave the same response as above.)

Round 2

Reviewer 1 Report

Authors have responded to all the issues appropriately. Based on the reviewers' suggestions, the paper has been revised. We recommend that this paper be accepted as is.

Reviewer 2 Report

This paper is now suitable for acceptance. Thanks